# Reinforcement Learning in a Birth and Death Process: Breaking the Dependence on the State Space

**Jonatha Anselmi**
jonatha.anselmi@inria.fr
Univ. Grenoble Alpes, Inria, CNRS, Grenoble INP, LIG, 38000 Grenoble, France.

**Bruno Gaujal**
bruno.gaujal@inria.fr
Univ. Grenoble Alpes, Inria, CNRS, Grenoble INP, LIG, 38000 Grenoble, France.

**Louis Sébastien Rebuffi**
louis-sebastien.rebuffi@univ-grenoble-alpes.fr
Univ. Grenoble Alpes, Inria, CNRS, Grenoble INP, LIG, 38000 Grenoble, France.

## Abstract

In this paper, we revisit the regret of undiscounted reinforcement learning in MDPs with a birth and death structure. Specifically, we consider a controlled queue with impatient jobs and the main objective is to optimize a trade-off between energy consumption and user-perceived performance. Within this setting, the *diameter $D$* of the MDP is $\Omega(S^S)$, where $S$ is the number of states. Therefore, the existing lower and upper bounds on the regret at time $T$, of order $O(\sqrt{DSAT})$ for MDPs with $S$ states and $A$ actions, may suggest that reinforcement learning is inefficient here. In our main result however, we exploit the structure of our MDPs to show that the regret of a slightly-tweaked version of the classical learning algorithm UCRL2 is in fact upper bounded by $\tilde{\mathcal{O}}(\sqrt{E_2AT})$ where $E_2$ is related to the weighted second moment of the stationary measure of a reference policy. Importantly, $E_2$ is bounded independently of $S$. Thus, our bound is asymptotically independent of the number of states and of the diameter. This result is based on a careful study of the number of visits performed by the learning algorithm to the states of the MDP, which is highly non-uniform.

## 1 Introduction

In the context of undiscounted reinforcement learning in Markov decision processes (MDPs), it has been shown in the seminal work [11] that the total regret of any learning algorithm with respect to an optimal policy is lower bounded by $\Omega(\sqrt{DSAT})$, where $S$ is the number of states, $A$ the number of actions, $T$ the time horizon and $D$ the *diameter* of the MDP. Roughly speaking, the diameter is the mean time to move from any state $s$ to any other state $s'$ within an appropriate policy. In the literature, several efforts have been dedicated to approach this lower bound. As a result, learning algorithms have been developed with a total regret of $\tilde{\mathcal{O}}(DS\sqrt{AT})$ in [11], $\tilde{\mathcal{O}}(D\sqrt{SAT})$ in [3] and even $\tilde{\mathcal{O}}(\sqrt{DSAT})$ according to [21, 25]. These results may give a sense of optimality since the lower bound is attained up to some universal constant. However, lower bounds are based on the minimax approach, which relies on the worst possible MDP with given $D$, $A$ and $S$. This means that when a reinforcement learning algorithm is used on a given MDP, one can expect a much better performance.

36th Conference on Neural Information Processing Systems (NeurIPS 2022).

One way to alleviate the minimax lower bound is to consider *structured reinforcement learning*, or equivalently MDPs with some specific structure. The exploitation of such structure may yield more efficient learning algorithms or tighter regret analyses of existing learning algorithms. In this context, a first example is to consider *factored* MDPs [6, 9], i.e., MDPs where the state space can be factored into a number of components; in this case, roughly speaking, $S = K^n$ where $n$ is the number of "factors" and $K$ is the number of states in each factor. The regret of learning algorithms in factored MDPs has been analyzed in [20, 17, 24, 14] and it is found that the $S$ term of existing upper bounds can be replaced by $nK$. A similar approach is used in [8] to learn the optimal policy in stochastic bandits with a regret that is logarithmic in the number of states. There is also a line of research works that exploit the parametric nature of MDPs. Inspired by parametric bandits, a $d$-linear additive model was introduced in [12], where it is shown that an optimistic modification of Least-Squares Value Iteration, see [15], achieves a regret over a finite horizon $H$ of $\tilde{\mathcal{O}}(\sqrt{d^3 H^3 T})$ where $d$ is the ambient dimension of the feature space (the number of unknown parameters). In this case, the regret does not depend on the number of states and actions and the diameter is replaced by the horizon. A discussion about the inapplicability of this approach to our case is postponed to Section 4.2.

**Learning in Queueing Systems.** The control of queueing systems is undoubtedly one of the main application areas of MDPs; see, e.g., [16, Chapters 1–3] and [13]. Within the rich literature of structured reinforcement learning however, few papers are dedicated to reinforcement learning in queueing systems, see [22, Section 5], and this motivates us to examine the total regret in this context. Typical control problems on queues have the following distinguishing characteristics:

1. *No discount.* Discounting costs or rewards is common practice in the reinforcement learning literature, especially in Q-learning algorithms [19]. However, in queues one is typically interested in optimizing with respect to the average cost.

2. *Large diameter.* Queueing systems are usually investigated under a drift condition that makes the system *stable*, i.e., positive recurrent. This condition implies that some states are hard to reach. In fact, for many queueing control problems, the diameter $D$ is exponential in the size of the state space. Even in the simple case of an M/M/1 queue with a finite buffer, or equivalently a birth–death process with a finite state space and constant birth and death rates, the diameter is exponential in the size of the state space.

3. *Structured transition matrices.* Queueing models describe how jobs join and leave queues, and this yields bounded state transitions. As a result, MDPs on queues have sparse and structured transition matrices.

The regret bounds discussed above and item 2 may suggest that the total regret of existing learning algorithm, when applied to queueing systems, is large. However, they often work well in practice and this bring us to consider the following question: *When the underlying MDP has the structure of a queueing system, do the diameter $D$ or the number of states $S$ actually play a role in the regret?*

**Our Contribution.** In this paper, we examine the previous question with respect to the class of control problems presented in [1]. Specifically, an infinite sequence of jobs joins a service system over time to receive some processing according to the first-come first-served scheduling rule; the system can buffer at most $S - 1$ jobs and in fact it corresponds to an M/M/1/S-1 queue. In addition, each job comes with a deadline constraint, and if a job is not completed before its deadline, then it becomes obsolete and is removed from the system. The controller chooses the server processing speed and the objective is to design a speed policy for the server that minimizes its average energy consumption plus an obsolescence cost per deadline miss. Although this may look quite specific, this problem captures the typical characteristics of a controlled queue: i) the transition matrix has the structure of a birth and death process with jump probabilities that are affine functions of the state and ii) the reward is linear in the state and convex in the action. For any MDP in this class, defined in full details in Section 3, we show that the diameter is $D = \Omega(S^{S-2})$; see Appendix B.3. Thus, without exploiting the particular structure of this MDP, the existing lower and upper bounds do not justify the reason why standard learning algorithms work efficiently here.

We provide a slight variation of the learning algorithm UCRL2, introduced in [11], and show in our main result that the resulting regret is upper bounded by $\tilde{\mathcal{O}}(\sqrt{E_2 AT})$ where $E_2$ is a term that depends on the stationary measure of a reference policy defined in Section 3.1. Importantly, $E_2$ does not depend on $S$. Thus, efficient reinforcement learning can be achieved independently of the number

of states by exploiting the stationary structure of the MDP. Let us provide some intuition about our result. First, one may think that any learning algorithm should visit each state a sufficient number of times, which justifies why the diameter of an MDP appears in existing regret analyses. However, this point of view does not take into account the fact that the value of an MDP is the scalar product of the reward and the stationary measure of the optimal policy. If this stationary measure is "highly non-uniform", then some states are rarely visited under the optimal policy and barely contribute to the value. In this case, we claim that the learner may not need to visit the rare states that often to get a good estimation of the value, and thus it may not need to pay for the diameter.

## 2 Reinforcement Learning Framework

We consider a unichain Markov Decision Process (MDP) $M = (\mathcal{S}, \mathcal{A}, P, r)$ in discrete time where $\mathcal{S}$ is the finite state space, $\mathcal{A}$ the finite action space, $P$ the transition probabilities and $r$ the expected reward function [16]. Let also $S := |\mathcal{S}|$ and $A := |\mathcal{A}|$ where $|\cdot|$ is the set cardinality operator. The model-based reinforcement learning problem consists in finding a *learning algorithm*, or learner, that chooses actions to maximize a cumulative reward over a finite time horizon $T$. At each time step $t \in \mathbb{N}$, the system is in state $s_t \in \mathcal{S}$ and the learner chooses an action $a_t \in \mathcal{A}$. When executing $a_t$, the learner receives a random reward $r_t(s_t, a_t)$ with mean $r(s_t, a_t)$ and the system moves, at time step $t+1$, to state $s'$ with probability $P(s'|s_t, a_t)$. The learning algorithm does not know the MDP $M$ except for the sets $\mathcal{S}$ and $\mathcal{A}$.

For simplicity, in the following we consider *weakly communicating* MDPs. Since we will be interested in the long-run average cost, this will let us remove the dependence on the initial state for several quantities of interest.

### 2.1 Undiscounted Regret

Given an MDP $M$, let $\Pi := \{\pi : \mathcal{S} \to \mathcal{A}\}$ denote the set of stationary and deterministic policies and let

$$\rho(M, \pi) := \lim_{T \to \infty} \frac{1}{T} \sum_{t=1}^{T} \mathbb{E}[r(s_t, \pi(s_t))] \tag{1}$$

denote the average reward induced by policy $\pi$. Since $M$ has finite state and action spaces, we notice that i) The limit in (1) always exists, ii) It does not depend on the initial state $s_0$ when $M$ is unichain [16] and iii) The restriction to stationary and deterministic policies is not a loss of optimality [16, Theorem 8.4.5].

Let also $\rho^* := \rho^*(M) := \max_{\pi \in \Pi} \rho(M, \pi)$ be the optimal average reward.

**Definition 2.1** (Regret). The *regret* at time $T$ of the learning algorithm $\mathbb{L}$ is

$$\text{Reg}(M, \mathbb{L}, T) := T\rho^*(M) - \sum_{t=1}^{T} r_t. \tag{2}$$

The regret 2 is a natural benchmark for evaluating the performance of a learning algorithm. In [11], a *universal* lower bound on $\text{Reg}(M, \mathbb{L}, T)$ has been developed in terms of the *diameter* of the underlying MDP.

**Definition 2.2** (Diameter of an MDP). Let $\pi : \mathcal{S} \to \mathcal{A}$ be a stationary policy of $M$ with initial state $s$. Let $T(s'|M, \pi, s) := \min\{t \geq 0 : s_t = s'|s_0 = s\}$ be the random variable for the first time step in which $s'$ is reached from $s$ under $\pi$. Then, we say that the *diameter* of $M$ is

$$D(M) := \max_{s \neq s'} \min_{\pi : \mathcal{S} \to \mathcal{A}} \mathbb{E}\left[T(s'|M, \pi, s)\right].$$

It should be clear that the diameter of an MDP can be large if there exist states that are hard to reach. Within the set of structured MDPs considered in this paper, this will be the case and we will show that $D = \Omega(S^{S-2})$. The following result shows that all learning algorithms have a regret that eventually increase with $\sqrt{D}$.

**Theorem 2.3** (Universal lower bound [11]). *For any learning algorithm $\mathbb{L}$, any natural numbers $S, A \geq 10$, $D \geq 20 \log_A S$, and $T \geq DSA$, there is an MDP $M$ with $S$ states, $A$ actions, and diameter $D$ such that for any initial state $s \in \mathcal{S}$,*

$$\mathbb{E}[\text{Reg}(M, \mathbb{L}, T)] \geq 0.015\sqrt{DSAT}. \tag{3}$$

In view of this result, the diameter of an MDP and its state space appear to be critical parameters for evaluating the performance of a learning algorithm.

## 2.2 The UCRL2 Algorithm

We now focus on UCRL2, a classical reinforcement learning algorithm introduced in [11] that is a variant of UCRL [2]. While more efficient algorithms have been proposed for the general case (see for example [3, 21]), we will show that UCRL2 already achieves a very low regret, namely $\tilde{\mathcal{O}}(\sqrt{AT})$, independent of $S$ so using more refined algorithms can only bring marginal gains.

UCRL2 is based on *episodes*. For each episode $k$, let $t_k$ denote its start time. For each state $s$ and action $a$, let $\nu_k(s, a)$ denote the number of visits of $(s, a)$ during episode $k$ and let $N_t(s, a) := \#\{\tau < t : s_\tau = s, a_\tau = a\}$ denote the number of visits of $(s, a)$ until timestep $t$. Let $\mathcal{M}_k$ be the confidence set of MDPs with transition probabilities $\tilde{p}$ and rewards $\tilde{r}$ that are "close" to the empirical MDP at episode $k$, $\hat{p}_k$ and $\hat{r}_k$, i.e., $\tilde{p}$ and $\tilde{r}$ satisfy

$$\forall (s, a), \quad |\tilde{r}(s, a) - \hat{r}_k(s, a)| \leq r_{\max} \sqrt{\frac{7 \log\left(2SAt_k/\delta\right)}{2 \max\left\{1, N_{t_k}(s, a)\right\}}} \tag{4}$$

$$\forall (s, a), \quad \|\tilde{p}(\cdot|s, a) - \hat{p}_k(\cdot|s, a)\|_1 \leq \sqrt{\frac{14S \log\left(2At_k/\delta\right)}{\max\{1, N_{t_k}(s, a)\}}} \tag{5}$$

With these quantities, a pseudocode for UCRL2 is given in Algorithm 1. We notice that UCRL2 relies on Extended Value Iteration (EVI), that is a variant of the celebrated Value Iteration (VI) algorithm [16]; for further details about EVI, we point the reader to [11, Section 3.1]. Let us comment on how UCRL2 works. There are three main steps. First, at the start of each episode, UCRL2

---

**Algorithm 1:** The UCRL2 algorithm.

**Input:** A confidence parameter $\delta \in (0, 1)$, $\mathcal{S}$ and $\mathcal{A}$.
**Output:** .
1 Set $t := 1$ and observe $s_1$
2 **for** *episodes $k = 1, 2, \ldots$* **do**
3      Compute the estimates $\hat{r}(s, a)$ and $\hat{p}_k(s'|s, a)$ as in (7).
4      Use "Extended Value Iteration" to find a policy $\tilde{\pi}_k$ and an optimistic MDP $\tilde{M}_k \in \mathcal{M}_k$ such that

$$\rho(\tilde{M}_k, \tilde{\pi}_k) \geq \max_{M' \in \mathcal{M}_k, \pi} \rho(M', \pi) - \frac{1}{\sqrt{t_k}} \tag{6}$$

5      **while** $\nu_k(s_t, \tilde{\pi}_k(s_t)) < \max\{1, N_{t_k}(s_t, \tilde{\pi}_k(s_t))\}$ **do**
6          Choose action $a_t = \tilde{\pi}_k(s_t)$, obtain reward $r_t$ and observe $s_{t+1}$;
7          $\nu_k(s_t, a_t) := \nu_k(s_t, a_t) + 1$;
8          $t := t + 1$;
9      **end**
10 **end**

---

computes the empirical estimates

$$\hat{r}_k(s, a) := \frac{\sum_{t=1}^{t_k-1} r_t \mathbb{1}_{\{s_t=s, a_t=a\}}}{\max\left\{1, N_{t_k}(s, a)\right\}}, \qquad \hat{p}_k(s'|s, a) := \frac{\sum_{t=1}^{t_k-1} \mathbb{1}_{\{s_t=s, a_t=a, s_{t+1}=s'\}}}{\max\left\{1, N_{t_k}(s, a)\right\}} \tag{7}$$

of the reward and probability transitions, respectively, where $\mathbb{1}_E$ is the indicator function of $E$. Then, it applies Extended Value Iteration (EVI) to find a policy $\tilde{\pi}_k$ and an optimistic MDP $\tilde{M}_k \in \mathcal{M}_k$ such that (6) holds true. Finally, it executes policy $\tilde{\pi}_k$ until it finds a state-action pair $(s, a)$ whose count within episode $k$ is greater than the corresponding state-action count prior to episode $k$.

# 3 Controlled Birth and Death Processes for Energy Minimization

Now, we focus on a specific class of MDPs that has been introduced in [1], which provides a rather general example of a controlled birth and death process with convex costs on the actions and linear rates. We will denote by $\mathcal{M}$ the set of MDPs with the structure described below. The MDPs in $\mathcal{M}$ have been proposed to represent a Dynamic Voltage and Frequency Scaling (DVFS) processor executing jobs with soft obsolescence deadlines. Here, jobs arrive according to a Poisson process with rate $\lambda \in [0, \lambda_{\max}]$ in a buffer of size $S - 1$. If the buffer is full and a job arrives, then the job is rejected. Each job has a deadline and a size, i.e., amount of work, which are exponentially distributed random variables with rates $\mu \in [0, \mu_{\max}]$ and one, respectively. Job deadlines and sizes are all independent random variables. If a job misses its deadline, which is a real time constraint activated at the moment of its arrival, it is removed from the queue without being served and a cost $C$ is paid. The processor serves jobs under any work-conserving scheduling discipline, e.g., first-come first-served, with a processing speed that belongs to the finite set $\{0, \ldots, A_{\max}\}$. The objective is to design a speed policy that minimizes the sum of the long term power dissipation and the cost induced by jobs missing their deadlines. When the processor works at speed $a \in \{0, \ldots, A_{\max}\}$, it processes $a$ units of work per second while its power dissipation is $w(a)$.

After uniformization, it is shown in [1] that this control problem can be modeled as an MDP in discrete time with a "birth-and-death" transition matrix of size $S$. Specifically, we have an MDP $M = (\mathcal{S}, \mathcal{A}, P, r)$ where $\mathcal{S} = \{0, \ldots, S - 1\}$, with $s \in \mathcal{S}$ representing the number of jobs in the system, and $\mathcal{A} = \{0, \ldots, A_{\max}\}$, with $a \in \mathcal{A}$ representing the processor speed. Then, the transition probabilities under policy $\pi$ are given by

$$P_{i,j}(\pi) = \begin{cases} \frac{1}{U}\lambda_i & \text{if } i < S - 1 \text{ and } j = i + 1 \\ \frac{1}{U}(\pi(i) + i\mu) & \text{if } i > 0 \text{ and } j = i - 1 \\ P_{ii} & \text{if } j = i \\ 0 & \text{otherwise,} \end{cases}$$

where $U := \lambda_{\max} + (S - 1)\mu_{\max} + A_{\max}$ is a uniformization constant, $P_{ii} = \frac{1}{U}(U - \lambda_i - \mu i - \pi(i))$ and $\lambda_i := \lambda\left(1 - \frac{i}{S-1}\right)$ is the *decaying* arrival rate. We have replaced the constant arrival rate $\lambda$ by a decaying arrival rate $\lambda_i$ because we want to learn an optimal policy that does not exploit the buffer size $S - 1$; see [1] for further details. For conciseness, Figure 1 displays the transition diagram of the Markov chain induced by policy $\pi$.

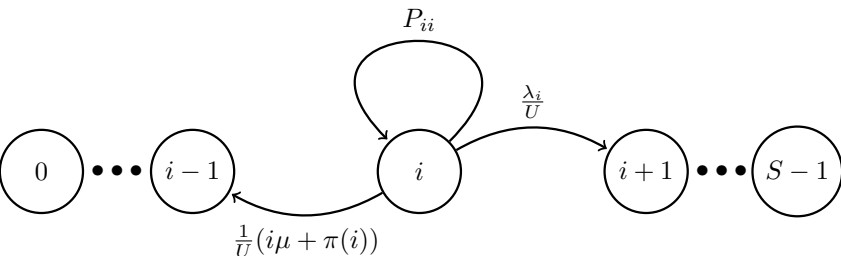

Figure 1: Transition diagram of the Markov chain induced by policy $\pi$ of an MDP in $\mathcal{M}$.

Finally, the reward is a combination of $C$, the constant cost due to a departing job missing its deadline and $w(a)$, an arbitrary convex function of $a$, giving the energy cost for using speed $a$. The immediate cost $c(s, a)$ in state $s$ under action $a$ is a random variable whose value is $w(a) + C$ with probability $i\mu/U$ (missed deadline) and $w(a)$ otherwise. To keep in line with the use of rewards instead of costs, we introduce a bound on the cost, $r_{\max} := C + w(A_{\max})$ so that the reward in state $s$ under action $a$ is a positive and bounded random variable given by

$$r(s, a) := r_{\max} - c(s, a). \tag{8}$$

As in Section 2.1, $\rho^*(M)$ is the optimal average cost and $\rho(M, \pi)$ is the average cost induced by policy $\pi$, where $\pi$ belongs to the set of deterministic and stationary policies $\Pi$. Since the underlying

Markov chain induced by any policy is ergodic, we observe that

$$\rho(M, \pi) = \sum_{s=0}^{S-1} \mathbb{E}[r(s, \pi(s))] m_s^\pi, \tag{9}$$

where $m^\pi$ is the stationary measure under policy $\pi$. In [1], it has been shown that the optimal policy is unique and will be denoted by $\pi^*$.

### 3.1 Properties of $\mathcal{M}$

In the following, we will use the "reference" (or bounding) policy $\pi^0(s) = 0$ for all $s \in \mathcal{S}$, which thus assigns speed 0 to all states. This policy provides a stochastic bound on all policies in the following sense. Let $s_t^\pi$ be the state under policy $\pi$ and let $\leq_{st}$ denote the *stochastic order* [18]; given two random variables $X$ and $Y$ on $\mathbb{R}_+$, we recall that $X \leq_{st} Y$ if $\mathbb{P}(X \geq s) \leq \mathbb{P}(Y \geq s)$ for all $s$.

**Lemma 3.1.** *Consider an MDP in $\mathcal{M}$. For all $t$ and policy $\pi \in \Pi$, $s_t^\pi \leq_{st} s_t^{\pi^0}$, provided that $s_0^\pi \leq_{st} s_0^{\pi^0}$.*

*Proof.* (sketch) The proof follows by a simple coupling argument between the two policies. Roughly speaking, each time the Markov chain under $\pi$ decreases from $s$ to $s-1$ because of the speed $\pi(s)$, it stays in state $s$ under policy $\pi^0$. $\qquad\square$

Therefore, $\mathbb{P}(s_t^\pi \geq s) \leq \mathbb{P}(s_t^{\pi^0} \geq s)$ for all $s$ and $t$, which also implies that the respective stationary measures are comparable, i.e., $\sum_{i=s}^{S-1} m_i^\pi \leq \sum_{i=s}^{S-1} m_i^{\pi^0}$.

Let us now consider $H(s)$, the *bias* at state $s$ of the optimal policy $\pi^*$, defined by

$$H(s) := \mathbb{E}_{\pi^*}\left[\sum_{t=1}^\infty \left( r\left(s_t^{\pi^*}, \pi^*(s_t^{\pi^*})\right) - \rho^*(M)\right) \mid s_0^{\pi^*} = s\right], \quad \forall 0 \leq s \leq S-1, \tag{10}$$

Let also $\partial H(s) := H(s) - H(s-1)$ be the local variation of the bias.

The following result was shown in [1, Lemma 3.8].

**Lemma 3.2.** *The local variation of the bias, $\partial H(s)$, is negative, decreasing in $s$, and bounded: $-\partial H(s) \leq \Delta(s)$ with $0 < \Delta(s) \leq C$ for all $1 \leq s \leq S-1$.*

Both $m^{\pi^0}$ and $\Delta$ will play a major role in our analysis of the regret.

### 3.2 Applying UCRL2 in $\mathcal{M}$

We assume that the bounds $\lambda_{\max}$ and $\mu_{\max}$ are fixed so that $r_{\max}$ is known to the learner. This is a classical assumption, often replaced by assuming that rewards live in $[0, 1]$.

In the remainder, we will apply UCRL2 over an MDP in $\mathcal{M}$ with a change in the confidence bounds to take into account the support of $P$. The confidence bounds in (4) (resp. (5)) are replaced by $r_{\max}\sqrt{\frac{2\log(2At_k)}{\max\{1, N_{t_k}(s,a)\}}}$ (resp. $\sqrt{\frac{8\log(2At_k)}{\max\{1, N_{t_k}(s,a)\}}}$). We also impose that the confidence set $\mathcal{M}_k$ only contains matrices with the same support as $P$. Removing $S$ in the confidence bounds does help to reduce the regret. However, by using existing analysis, this only removes a factor $\sqrt{S}$ in the regret bound (for example, see [4]).

Finally, note that UCRL2 does not benefit from the parametric nature of the MDPs in $\mathcal{M}$, which is essentially defined by three parameters ($\lambda$, $\mu$ and $C$) and the real convex function $w(\cdot)$.

## 4 Regret of UCRL2 on $\mathcal{M}$

Our objective is to develop an upper bound on the regret of the learning algorithm UCRL2 when applied to MDPs in our class $\mathcal{M}$. The driving idea is to construct a bound that exploits the structure of the stationary measure of all policies, as they all make some states hard to reach, and to control the number of visits to these states to get a new type of bound.

### 4.1 Main Result

The following theorem gives an upper bound on the regret that does not depend on the classical parameters such as the size of the state space nor on global quantities such as the diameter of the MDP nor the span of the bias of some policy. Instead, the regret bound below mainly depends on the weighted second moment of the stationary measure of the reference policy $\pi^0$, which is bounded independently of the size of the state space.

We consider the policy $\pi^{\max}$ such that $\pi^{\max}(s) = A_{\max}$ for all $s$ and $m^{\max}$ its stationary measure.

Let us also recall that $m^{\pi^0}$ is the stationary measure of the Markov chain under policy $\pi^0(s) = 0$ for all $s$ and that $\Delta : \mathcal{S} \to \mathbb{R}^+$ is a function bounding the local variations of the optimal bias. Let $E_2 := F \, \mathbb{E}_{m^{\pi^0}} \left[ (\Delta + r_{\max})^2 \cdot f \right]$ with $f : s \mapsto \frac{\max\{1, s(s-1)\}}{(\Delta(s) + r_{\max})^2}$ and $F := \sum_{s \in \mathcal{S}} f(s)^{-1}$. Here, $E_2$ is closely related to the second moment of the measure $m^{\pi^0}$ weighted by the bias variations and the maximal reward.

**Theorem 4.1.** *Let* $M \in \mathcal{M}$. *Define* $Q_{\max} := \left( \frac{10D}{m^{\max}(S-1)} \right)^2 \log \left( \left( \frac{10D}{m^{\max}(S-1)} \right)^4 \right)$.

$$\mathbb{E}\left[ \mathrm{Reg}(M, \mathrm{UCRL2}, T) \right] \leq 19 \sqrt{E_2 A T \log (2AT)}$$
$$+ 97 r_{\max} D^2 S A \max\{ Q_{\max}, T^{1/4} \} \log^2 (2AT). \quad (11)$$

*Here,* $E_2 \leq 12 r_{\max}^2 \left( 1 + \frac{\lambda^2}{\mu^2} \right)$, *so that the regret satisfies*

$$\mathbb{E}\left[ \mathrm{Reg}(M, \mathrm{UCRL2}, T) \right] = \mathcal{O} \left( r_{\max} \sqrt{AT \left( 1 + \frac{\lambda^2}{\mu^2} \right) \log (AT)} \right).$$

Before giving a sketch of the proof, let us comment on the bound (11). Although the first term is of order $\sqrt{T}$ with a multiplicative constant independent of $S$ - as desired - the second term, of order $T^{1/4}$, contains very large terms. Its interest however, lies in the novel approach used in the proof that uses the stationary behavior of the algorithm.

### 4.2 Comparison with Other Bounds

Let us compare our upper bound with the ones existing in the literature, as we claim that ours is of a different nature.

First, let us compare with the bound given in [11], which states that with probability $1 - \delta$, $\mathrm{Reg}(M, T) \leq 34 D S \sqrt{AT \log \left( \frac{T}{\delta} \right)}$ for any $T > 1$. For any $M \in \mathcal{M}$, the diameter grows as $S^S$ (see Appendix B.3), thus this bound is very loose here. More recent works have improved this bound by replacing the term $D$ by the local diameter of the MDP [5]. In Appendix B.3, we show that the local diameter grows again as $S^S$ for any $M \in \mathcal{M}$, and thus these results do not yield significant improvements. Other papers show that the diameter can be replaced by the span of the bias, see [7, 25]. This has a big impact because the span of the bias, for any $M \in \mathcal{M}$, is linear in $S$ (instead of $S^S$ for the diameter); see Appendix B.3. However, this is still not as good as the bound given in Theorem 4.1, which is independent of $S$.

Now, let us compare with existing bounds for *parametric* MDPs, as mentioned in the introduction. The $d$-linear additive model, $d < S$, introduced in [12] assumes that $P(\cdot|s, a) = \langle \phi(s, a), \theta(\cdot) \rangle$, where $\phi(s, a)$ is a known feature mapping and $\theta$ is an unknown measure on $\mathbb{R}^d$. This form of $P(\cdot|s, a)$ implies that the transition kernel is of rank $d$. Unfortunately, this property does not hold true in birth and death processes. In fact, the kernel of any $M \in \mathcal{M}$ has almost *full* rank under all policies. The *linear mixture model* introduced in [26] assumes instead that $P(s'|s, a) = \langle \phi(s'|s, a), \theta \rangle$, $\theta \in \mathbb{R}^d$. This is more adapted to our case, which can be (almost) seen as a linear mixture model of dimension $d = 3$. The bound on the discounted regret of the algorithm proposed in [26] is $\mathrm{Reg}(M, T) \leq d\sqrt{T}/(1 - \gamma)^2$ where $\gamma$ is a discount factor. In contrast to our work, this regret analysis holds for *discounted* problems, where we remark that both the diameter and the span are irrelevant. On the other hand, both are replaced by a term of the form $(1 - \gamma)^{-2}$, which implies

that the previous bound grows to infinity as $\gamma \uparrow 1$. More recently, a regret bound of $O(d\sqrt{DT})$ has been proven in [23] in the undiscounted case, that is the case considered in our work. However, the algorithm presented in that reference highly depends on the diameter and is unsuitable for MDPs with a birth and death structure.

Finally, our bound depends on the second moment of the stationary measure of a reference policy, i.e., $E_2$, which can be bounded independently of the state space size. This is structurally different from the ones existing in the literature. We believe that this structure holds as well in a class of MDPs much larger than $\mathcal{M}$. In particular, if $m$ is the stationary measure of some bounding/reference policy, and if the critical quantity $\mathbb{E}_m [\Delta \cdot f]$ is small for a well chosen function $f$, then the regret of a learning algorithm navigating the MDP should also be small. A deeper analysis is left as future work.

## 4.3 Sketch of the Proof

Our proof for Theorem 4.1 is technical and is provided in the supplementary material. In this section, we present the main ideas and its general structure. It initially relies on the regret analysis of UCRL2 developed in [11], and the differences are highlighted below. First, we consider the mean rewards and split the regret into episodes to separately treat the cases where the true MDP is in the confidence set of optimistic MDPs $\mathcal{M}_k$ or not. Thus, let $R_k := \sum_{s,a} \nu_k(s,a)(\rho^* - \overline{r}(s,a))$ denote the regret in episode $k$. This split can be written:

$$\mathbb{E}\left[\text{Reg}(M,T)\right] \leq \mathbb{E}\left[R_{\text{in}}\right] + \mathbb{E}\left[R_{\text{out}}\right],$$

where $R_{\text{in}} := \sum_k R_k \mathbb{1}_{M \in \mathcal{M}_k}$ and $R_{\text{out}} := \sum_k R_k \mathbb{1}_{M \notin \mathcal{M}_k}$.

To control $R_{\text{out}}$, we use, as in [11], the stopping criterion and the confidence bounds. This gives $\mathbb{E}\left[R_{\text{out}}\right] \leq r_{\max}S$, so that the regret due to episodes where the confidence regions fail will be negligible next to the main terms. Then, when the true MDP belongs to the confidence region, we use the properties of Extended Value Iteration (EVI) to decompose $R_{\text{in}}$ into

$$\underbrace{\sum_{k,s,a} \nu_k(s,a)(\tilde{r}_k - \overline{r}(s,a))}_{R_{\text{rewards}}} + \underbrace{\sum_k \mathbf{v}_k \left(\tilde{\mathbf{P}}_k - \mathbf{I}\right) \tilde{\mathbf{h}}_k}_{R_{\text{bias}}} + \underbrace{\sum_k \mathbf{v}_k \left(\tilde{\mathbf{P}}_k - \mathbf{I}\right) \mathbf{d}_k + 2r_{\max} \sum_{k,s,a} \frac{\nu_k(s,a)}{\sqrt{t_k}}}_{R_{\text{EVI}}},$$

where $\mathbf{v}_k$ is the vector of the state-action counts $\nu_k$'s, $\tilde{\mathbf{P}}_k$ and $\tilde{\mathbf{h}}_k$ are respectively the transition matrix and the bias in $\tilde{M}_k$ under policy $\tilde{\pi}_k$, and $\mathbf{d}_k$ is the profile difference between the last step of EVI and the bias (see Appendix A.3).

We now show how to handle $R_{\text{rewards}}$, $R_{\text{EVI}}$, $R_{\text{bias}}$. First, we deal with the terms that do not involve the bias. Using the confidence bound on the rewards (see Appendix A.3.1:

$$R_{\text{rewards}} \leq r_{\max} 2\sqrt{2\log(2AT)} \sum_k \sum_{s,a} \frac{\nu_k(s,a)}{\sqrt{\max\left\{1, N_{t_k}(s,a)\right\}}}. \tag{12}$$

Now, let us consider $R_{\text{EVI}}$. Since $\mathbf{d}_k$ becomes arbitrarily small after enough iterations of EVI (see Appendix A.1), for $T \geq \frac{e^8}{2AT}$, we get

$$R_{\text{EVI}} \leq r_{\max} 2\sqrt{2\log(2AT)} \sum_k \sum_{s,a} \frac{\nu_k(s,a)}{\sqrt{\max\left\{1, N_{t_k}(s,a)\right\}}}. \tag{13}$$

The analysis of $R_{\text{bias}}$ is different from the one in [11]: While in [11] the bias is directly bounded by the diameter, we can use the variations of the bias to control the regret more precisely. Using $\mathbf{P}_k$, i.e., the transitions in the true MDP under $\tilde{\pi}_k$, $R_{\text{bias}}$ is further decomposed into:

$$\underbrace{\sum_k \mathbf{v}_k \left(\tilde{\mathbf{P}}_k - \mathbf{P}_k\right) \mathbf{h}^*}_{R_{\text{trans}}} + \underbrace{\sum_k \mathbf{v}_k \left(\tilde{\mathbf{P}}_k - \mathbf{P}_k\right) \left(\tilde{\mathbf{h}}_k - \mathbf{h}^*\right)}_{R_{\text{diff}}} + \underbrace{\sum_k \mathbf{v}_k \left(\mathbf{P}_k - \mathbf{I}\right) \tilde{\mathbf{h}}_k}_{R_{\text{ep}}}.$$

The term $R_{\text{ep}}$ can be treated in a similar manner as in [11] by bounding the bias terms with the diameter to apply an Azuma-Hoeffding inequality (see Appendix A.3.5). Here, we obtain

$$\mathbb{E}\left[R_{\text{ep}}\right] \leq SAD\, r_{\max} \log_2\left(\frac{8T}{SA}\right).$$

Next, we show in A.3.2 that $R_{\text{diff}}$ does not contribute to the main term of the regret. This is one of the hard point in our proof. First, linear algebra techniques are used to bound $||\tilde{\mathbf{h}}_k - \mathbf{h}^*||_\infty$ by $D(2r_{\max}D||\tilde{\mathbf{P}}_k - \mathbf{P}^*||_\infty + ||\tilde{\mathbf{r}}_k - \mathbf{r}^*||_\infty)$. Each norm is then bounded using Hoeffding inequality. This introduces the special quantity $N_{t_k}(x_k, \pi_k(x_k))$ that yields to the worst confidence bound in episode $k$. Then, an adaptation of McDiarmid's inequality to Markov chains is used to show that $N_{t_k}(x_k, \pi_k(x_k)) \geq (t_{k+1} - t_k)m^{\max}(S - 1)/2$ with high probability, where $m^{\max}(S - 1)$ is the stationary measure of state $S - 1$ under the uniform policy $\pi^{\max}(s) = A_{\max}$. This eventually implies that

$$\mathbb{E}[R_{\text{diff}}] \leq 96r_{\max}D^2 SA \max\{Q_{\max}, T^{1/4}\}\log^2(2AT),$$

where $Q_{\max} := \left(\frac{10D}{m^{\max}(S-1)}\right)^2 \log\left(\left(\frac{10D}{m^{\max}(S-1)}\right)^4\right)$.

Then, to deal with the main term $R_{\text{trans}}$, we exploit the optimal bias. The unit vector being in the kernel of $\tilde{\mathbf{P}}_k - \mathbf{P}_k$, we can rewrite:

$$R_{\text{trans}} = \sum_k \sum_s \sum_{s'} \nu_k(s, \tilde{\pi}_k(s)) \cdot (\tilde{p}_k(s'|s, \tilde{\pi}_k(s)) - p(s'|s, \tilde{\pi}_k(s))) \cdot (h^*(s') - h^*(s))$$

and, thus, using the confidence bound and the bounded variations of the bias,

$$R_{\text{trans}} \leq 4\sqrt{2\log(2AT)} \sum_k \sum_{s,a} \frac{\Delta(s)\nu_k(s,a)}{\sqrt{\max\{1, N_{t_k}(s,a)\}}}.$$

We can now aggregate $R_{\text{trans}}$, $R_{\text{rewards}}$ and $R_{\text{EVI}}$ to compute the main term of the regret (see Appendix A.3.4). Here, the key ingredient is to bound

$$\sum_{k,s,a} \frac{(\Delta(s) + r_{\max})\nu_k(s,a)}{\sqrt{\max\{1, N_{t_k}(s,a)\}}}$$

independently of $S$. This is the second main difference with [11]. Instead of exploring the MDP uniformly, we know that the algorithm will mostly visit the first states of the MDP, regardless of the chosen policy. As shown in [11], for a fixed state $s$:

$$\mathbb{E}\left[\sum_a \sum_k \frac{\nu_k(s,a)}{\sqrt{\max\{1, N_{t_k}(s,a)\}}}\right] \leq 3\sqrt{\mathbb{E}[N_T(s)]A}.$$

Now, instead of summing over the states, we can use properties of stochastic ordering to compare the mean number of visits of a state with the probability measure $m^{\pi^0}$; here, we strongly rely on the birth and death structure of the MDPs in $\mathcal{M}$. For any non-negative non-decreasing function $f : \mathcal{S} \to \mathbb{R}^+$, we obtain

$$\mathbb{E}\left[\sum_{s\geq 0} f(s)N_t(s)\right] \leq t\sum_{s\geq 0} f(s)m^{\pi^0}(s). \tag{14}$$

Let us choose $f : s \mapsto \frac{\max\{1, s(s-1)\}}{(\Delta(s) + r_{\max})^2}$ and let $F := \sum_s f(s)^{-1} \leq 3(C + r_{\max})^2$. Let also $E_2 := F\mathbb{E}_{m^{\pi^0}}\left[(\Delta + r_{\max})^2 \cdot f\right]$. Then,

$$\mathbb{E}\left[\sum_k \sum_{s,a} \frac{(\Delta(s) + r_{\max})\nu_k(s,a)}{\sqrt{\max\{1, N_{t_k}(s,a)\}}}\right] \leq 3\sqrt{E_2 AT}.$$

In Appendix A.3.4, we further show that $E_2 \leq 3(C + r_{\max})^2\left(1 + \frac{\lambda^2}{\mu^2}\right)$. Therefore, for the three main terms, we obtain

$$\mathbb{E}[R_{\text{trans}} + R_{\text{rewards}} + R_{\text{EVI}}] \leq 19\sqrt{E_2 AT \log(2AT)} \tag{15}$$

and we conclude our proof by combining all of these terms.

## 5 Conclusions

For learning in a class of birth and death processes, we have shown that exploiting the stationary measure in the analysis of classical learning algorithms yields a $K\sqrt{T}$ regret, where $K$ only depends on the stationary measure of the system under a well chosen policy. Thus, the dependence on the size of the state space as well as on the diameter of the MDP or its span disappears. We believe that this type of results can be generalized to other cases such as optimal routing, admission control and allocation problems in queuing systems, as the stationary distribution under all policies is uneven between the states.

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
