# OpenReview forum: "Reinforcement Learning in a Birth and Death Process: Breaking the Dependence on the State Space"
_NeurIPS.cc/2022/Conference — NeurIPS 2022 Accept_

### Official Review · Reviewer_Sx4m · 2022-07-11

**Rating:** 5
**Confidence:** 4
**Soundness:** 3 good
**Presentation:** 3 good
**Contribution:** 2 fair

**Summary:**

This work studies RL on a special class of MDPs called the birth and death process (BDP for short). The authors claim that by applying the UCRL2 algorithm to BDP, a state-independent regret can be obtained, which improves the existing best-known regret bounds that depend on the state space cardinality explicitly.

**Questions:**

+ Can the authors highlight their theoretical contributions?
+ Can the authors compare their work and Wu et al., 2022?

**Limitations:**

Yes.

**Strengths And Weaknesses:**

Strengths:

+ The presentation is clear.
+ The theoretical results are technically sound.

Weaknesses:

+ The importance of this work remains unclear. This work does not propose either a new algorithm or a new proof approach. It seems that the only difference between their algorithm and proof and those of [10] is some more refined analysis which depends on the quantity $r_{max}$, as they showed in (12, 13). However, the difficulty to analyze it is unknown, and I feel it is not important enough to be the main contribution of a NeurIPS paper.

+ The comparison of some highly related works lacks. In Section 4.2, the authors claim that BDPs can be regarded as a special class of linear mixture MDPs, and existing algorithms for linear mixture MDPs are for discounted MDPs and can not be directly applied to BDPs . It is not true since Wu et al., 2022 have already studied linear mixture MDPs with average rewards. Given the existing results, it is hard to judge the contribution of this work.


Wu, Y., Zhou, D., & Gu, Q. (2022, May). Nearly minimax optimal regret for learning infinite-horizon average-reward mdps with linear function approximation. In International Conference on Artificial Intelligence and Statistics (pp. 3883-3913). PMLR.

---

> ### Author Response · Authors · 2022-07-28
> **Answers to Reviewer Sx4m**
>
> Answers:
>
> We answer to both questions 1 and 2.
>
> We would like to thank Rev. 4 for pointing out the work of Wu et al. (that appeared  during the submission period of NeurIPS). We will cite this paper in Section 4.2 together with the discounted case. However, their results are different from ours. First, they use a specific learning algorithm that knows the structure of the mixture linear model, while our algorithm is essentially oblivious to the structure (albeit for the bonus on the transition matrix). Second, and most importantly, their bound is in $O(d\sqrt{DT})$ and thus depends on the diameter $D$ of the MDP -- we recall that our main theoretical contribution is to have an upper bound independent of the diameter. Applying their bound in our case would give a regret bound in $O(3\sqrt{S^S T})$ because $D = S^S$ here-- we recall that in contrast our bound does not depend on $S$. The main goal of our paper is precisely to show that, for this type of B \& D processes, one can find a regret bound that does not depend on the diameter but instead on a bounded function of the stationary measure of the system.
>
> The fact that $r_\max$ (the maximum reward) appears in the theorem is a small generalization of the classical convention taken by almost all papers in this domain (as well as in Wu et al., see Section 3)  that assume $r_\max =1$. Again, our main contribution is finding a bound asymptotically independent of the size of the state space, of the diameter and even of the span of the bias, that usually appear in regret bounds.
>
> At the mathematical level, while we follow the same classical decomposition of the regret as in [10] and other works, our proof also relies on McDiarmid's inequality for Markov chains, while existing approaches only rely on concentration inequalities. In addition, we use coupling and monotonicity arguments that come from queueing theory. All these arguments imply the introduction of a new term in the regret bound that depends on the stationary measure. To the best of our knowledge, this has not been done before.

---

> > ### Comment · Reviewer_Sx4m · 2022-08-06
> > **Reply**
> >
> > Thanks for your reply. Now I get that the main contribution is to provide a new problem-dependent measure, at least for the Birth and Death process. I have increased my score from 3 to 5.

---

### Official Review · Reviewer_NyfN · 2022-07-11

**Rating:** 5
**Confidence:** 2
**Soundness:** 3 good
**Presentation:** 2 fair
**Contribution:** 3 good

**Summary:**

This paper proposed a new undiscounted MDP setting with a birth and death structure. Then the authors modified the classical UCRL2 algorithm to achieve an $\widetilde{O}(\sqrt{E_2AT})$, which has no dependence on the number of the states and the diameter. This is achieved by leveraging the special structure of the queueing system.

**Questions:**

- I didn't see why $E_2$ can be interpreted as the second moment from the definition in Line 225. Can the authors elaborate more on this point?
- What is the upper bound on $E_2$?

**Limitations:**

I am not aware of any potential negative societal impact

**Strengths And Weaknesses:**

Strengths:
- This paper formulated queueing as an MDP with birth and death structure and proposed an efficient algorithm, which is important as the queueing system is widely applied in many different areas.
- The regret bound has no dependence on the number of states or the diameter thanks to the special structure. On the other hand, it depends on $E_2$, which is a weighted second moment of the stationary measure of a reference policy. This result is interesting as it is specialized to the problem structure.

Weakness:
- The writing part can be improved. I didn't understand why $E_2$ can be interpreted as the second moment and why dependence on the number of states or the diameter can be removed. As a result, I am not certain about the correctness o this paper.

---

> ### Author Response · Authors · 2022-07-28
> **Answers to Reviewer NyfN**
>
> Answers:
> 1. The expectation in line 225 is equal to $E_{m^{\pi^0}}[\max\{1,s(s-1)\}]$ where $s$ is the random state of the Markov chain. We have $E_{m^{\pi^0}}[\max\{1,s(s-1)\}]=m^{\pi^0}(0) + E_{m^{\pi^0}}[s^2] - \mathbb{E}_{m^{\pi^0}}[s]$ and here we can see the relationship with the second moment. We agree that our statement in lines 225-227 "$E_2$ can be seen as the second moment of the measure $m^{\pi^0}$ weighted by the bias variations and the maximal reward." is not precise. We will replace it with "$E_2$ is closely related to the second moment of the measure $m^{\pi^0}$ weighted by the bias variations and the maximal reward". A similar modification will also be made in the abstract.
> 2. An upper bound on $E_2$ is given in Theorem 4.1 (Line 229) and is derived in Appendix A.3.4. The important point in this bound is that it is independent of the size of the state space.

---

> > ### Comment · Reviewer_NyfN · 2022-08-06
> > **Thanks for your reply**
> >
> > Thanks for your reply! After reading other reviews and replies, I would like to keep my score.

---

> ### Author Response · Authors · 2022-07-29
> **Additional comment for Reviewer NyfN**
>
> Although this was not a direct question from the reviewer, we would like to comment on the fact that the bound on the regret does not depend on the state space, which indeed looks counter-intuitive.
>
> First we should point out that our bound does depend on $S$, but only in the non-dominant terms. As for the dominant term, a non-rigorous explanation is that this dominant term only depends on the average behavior of the learning algorithm. This is bounded by the average behavior of the Markov process under measure $m^0$. Under $m^0$, the large states are very rarely visited and on average, this process remains bounded, independently of $S$.

---

### Official Review · Reviewer_SMUj · 2022-07-12

**Rating:** 7
**Confidence:** 4
**Soundness:** 4 excellent
**Presentation:** 4 excellent
**Contribution:** 4 excellent

**Summary:**

This paper studies RL in undiscounted MDPs. The authors break the minimax lower bound which depends on the MDP diameter (shown to be as large as $\Omega(S^S)$) by restricting the MDPs to a structured class of birth-death processes and showing that a tweaked version of the classic UCRL2 algorithm achieves an upper bound independent of the diameter and state size. This bound is obtained by carefully analyzing the stationary measure for this restricted MDP class and exploiting its high non-uniformity.

**Questions:**

Why did you consider UCRL instead of UCB?

**Limitations:**

The work is theoretical.

**Strengths And Weaknesses:**

**Strengths**

- The problem studied in this paper is related to finding structures in RL that facilitate efficient learning is an important topic. Many existing works in finite-sample analysis of RL have focused on algorithms that are nearly optimal in a minimax sense, where the optimality is measure w.r.t. the most difficult MDP. However, this does not explain why RL works well in certain environments. Studying structed RL is important to bridge the gap between theory and practice and this paper contributes to this line of research.
- The paper considers a specific structure (birth death) related to queueing systems, which itself has important applications and seems to be motivated by practical implications. Moreover, this structure is different from usual structures considered in RL theory literature such as low rank MDPs. Although the minimax lower bound depends on diameter (which is large for birth death process), the paper carefully analyzes regret for this restricted class of MDPs to identify whether there is a diameter or state size dependency. The authors show that a variant of UCRL2 achieves a regret independent of these parameters.
- The technical contributions are insightful and may inspire future work. For instance, although UCRL2 does not exploit the parameteric form of MDP, it still achieves a regret independent of diameter and state size. Another technique used in the regret guarantee is exploiting the highly non-uniform stationary measure in the restricted MDP class.
- The paper is very well-written and thorough comparison with prior related theoretical results and their implications is provided.

**Weaknesses**

- The structure considered in this work may be more limited compared to other structures in prior works such as factored MDPs, which may limit the significance and impact of this work.
- The paper shows that a variation of UCRL2 achieves a regret independent of diameter and state size. It remains unclear whether the tweak was necessary, the regret is optimal, and other popular methods such as UCB also enjoy similar guarantees. Additional theoretical and/or empirical results can strengthen the work.

---

> ### Author Response · Authors · 2022-07-28
> **Answers to Reviewer SMUj**
>
> Answers:
>
> The  UCB (Upper Confidence Bound) algorithm only applies to stochastic bandits (without states), while UCRL can actually be seen as the adaptation of UCB to MDPs. So UCRL is a natural choice here.

---

### Official Review · Reviewer_sAR5 · 2022-07-15

**Rating:** 6
**Confidence:** 4
**Soundness:** 3 good
**Presentation:** 3 good
**Contribution:** 3 good

**Summary:**

This paper revisits the regret of undiscounted reinforcement learning in MDPs with a birth and death structure. Specifically, they consider a controlled queue with impatient jobs and the main objective is to optimize a trade-off between energy consumption and user-perceived performance. They exploit the structure of the MDP and give a tighter upper bound on the regret of a slightly-tweaked version of the classical learning algorithm UCRL2.  Their bound is asymptotically independent of the number of states and of the diameter.

**Questions:**

1. Are there any lower bounds on the setting considered in this paper?
2. Can the authors conduct numerical experiments to shoe the effectiveness of the method?

**Limitations:**

The authors have adequately addressed the limitations and potential negative societal impact of their work.

**Strengths And Weaknesses:**

Strengths:
Over all, this paper studies a special case of MDP with a birth and death structure. The MDP case is commonly used in practice, and is worth studying. In theoretical analysis, the authors studies a slightly-tweaked version of the classical learning algorithm UCRL2, which is relatively simple and widely used. By exploiting the structure of the MDP, they provide a much tighter upper bound on the performance of UCRL2, which, surprisingly, is independent of the number of states, and improves previous naive results which exponentially depends on S. This paper is interesting, and is important to RL theory.

Weaknesses:
1. There are no corresponding lower bounds, so we do not know whether the upper bound is tight or not.
2. It will be better if the authors can provide experiment results to support their algorithm.

---

> ### Author Response · Authors · 2022-07-28
> **Answers to Reviewer sAR5**
>
> Answers:
> 1. To the best of our knowledge, no lower bounds exist within the considered setting. In fact, lower bounds are based on the minimax approach and they depend on the worst possible MDP with given D, A and S. The model considered in our paper is highly constrained and lower bounds are not as appropriate in this context.
>
> 2. The bound presented here is of asymptotic nature (non-dominant terms in $T^{1/4}$ can be very large with a small $T$ and this hides the independence on the state space in the dominant term). Furthermore, even in the general case, the numerical experiments on a single given MDP do not usually behave as $\sqrt{T}$: The regret can grow as $\frac{1}{g}\log(T)$ (where $g$ is the sub-optimality gap of this MDP) or even become constant over very long periods. A numerical investigation can hardly overcome these issues.

---

### Meta-Review · Area_Chair_v635 · 2022-08-26

**Recommendation:** Accept
**Confidence:** Less certain

**Metareview:**

This paper studies reinforcement learning in a restrictive set of MDPs. It showed that a tweaked version of the classic UCRL2 algorithm achieves an upper bound independent of the diameter and state size. This bound is obtained by carefully analyzing the stationary measure for this restricted MDP class and exploiting its high non-uniformity. It is an important problem to identify structure in the MDP that enables efficient learning.

**Award:**

No

---

### Decision · Program_Chairs · 2022-09-14

Accept